# Technical Limits in Circularity for Plastic Packages

**Marieke T. Brouwer** [1], **Eggo U. Thoden van Velzen** [1,*], **Kim Ragaert** [2] **and Roland ten Klooster** [3]

1 Wageningen Food & Biobased Research, Wageningen University & Research, Bornse Weilanden 9, 6709 WG Wageningen, The Netherlands; marieke.brouwer@wur.nl
2 Centre for Polymer and Material Technologies, Department of Materials, Textiles and Chemical Engineering, Ghent University, Technologiepark 130, 9052 Zwijnaarde, Belgium; kim.ragaert@ugent.be
3 Faculty of Engineering Technology, University of Twente, De Horst 2, 7522 LW Enschede, The Netherlands; r.tenklooster@utwente.nl
* Correspondence: ulphard.thodenvanvelzen@wur.nl

**Abstract:** The current Dutch recycling value chain for plastic packaging waste (PPW) has not reached its full circularity potential, as is apparent from two Circular Performance Indicators (CPIs): net packaging recycling rate and average polymer purity of the recycled plastics. The performance of the recycling value chain can be optimised at four stages: packaging design, collection, sorting, and recycling. This study explores the maximally achievable performance of a circular PPW recycling value chain, in case all stakeholders would implement the required radical improvement measures in a concerted action. The effects of the measures were modelled with material flow analysis. For such a utopic scenario, a net plastic packaging recycling rate of 72% can be attained and the produced recycled plastics will have an average polymeric purity of 97%. This is substantially more than the net packaging recycling rate of 37% for 2017 and will exceed the EU target of 50% for 2025. In such an ideal circular value chain more recycled plastics are produced for more demanding applications, such as food packaging, compared to the current recycling value chain. However, all stakeholders would need to implement drastic and coordinated changes, signifying unprecedented investments, to achieve this optimal circular PPW recycling value chain.

**Keywords:** plastic packaging waste; recycling; recycling targets; polymer purity; quality of recycled plastics; limits

## 1. Introduction

The European Union strives towards a circular economy for plastics to mitigate the environmental impacts of plastic waste. Plastic packaging waste (PPW) is a priority since roughly 40% of the plastics are used in packages and plastic packages make up 60% of the plastic waste [1]. Since the 1990s, member states have established multiple collection and recycling systems for PPW that generate various qualities of recycled plastics. Several recycling systems have been thoroughly analysed and share common features [2–7]. The homogenous post-industrial plastic packaging waste (PI-PPW) flows are recycled into relatively pure recycled plastics that can be applied in related packaging and non-packaging applications. The more heterogeneous PI-PPW and post-consumer plastic packaging waste (PC-PPW) are connected to a network of sorting and recycling facilities and generate various types of recycled plastics that are often blends of polymers and are often only applicable in non-food packaging and non-packaging applications. Currently, only a small share of food packages are recycled into food-grade recycled plastics, which is to a large degree determined by legislative constraints [8]. Well-known are the post-consumer polyethene terephthalate (PET) beverage bottles which are recycled to food-grade PET bottles and trays [9,10]. Much smaller recycling activities are the British separately collected high-density polyethene (HDPE) milk jugs to food-grade recycled polyethene (PE) [11] and

roughly a dozen reusable polypropylene (PP) and PE food crates, which are recycled to food-grade PE and PP [12,13]. Only a fraction of the recycled PE, PP, and Film is used in packaging (rPE flasks, rPP boxes and crates and rLDPE film) and other high demand applications. The majority of these recycled plastics are still used in low demand non-packaging applications. Moreover, one of the largest recycling products (mixed plastics) is used in fairly bulky applications, such as garden furniture, fence posts, pallets, etc., [3,14]. The current overall mismatch in the (both regulatory and technical) qualities of recycled plastics offered and the qualities the packaging industry want to procure [15], retards the transition towards a circular economy. Several technological improvement measures are known to improve the circular performance of the recycling system to some extent. Design-for-recycling measures by the packaging industries, in general, affect the polymeric purity of the recycled plastics and hence their applicability [16]. In contrast, more intensive collection and mechanical recovery of plastics from mixed municipal solid waste (MSW) can increase the net plastic packaging recycling rates [3,17]. On top of that, sorting and recycling facilities can also contribute to generating slightly more and more pure recycled plastics with investments in new improved separation technologies [18,19]. This raises the question what the theoretical limit of a PPW recycling value chain is. In other words, in case all the stakeholders would co-operate, what level of circularity could maximally be achieved when all the improvement measures are implemented in a concerted action. Although this is a utopic scenario, this theoretical exercise does provide valuable perspectives on the limits of what could potentially be achieved.

Circular indicators assess the level of circularity that a product, company or collection and recycling network has achieved. Hundreds of these indicators have previously been proposed and used [20–22]. Specific for PPW, simple recycling rates are insufficient since these do not account for the quality of the recycled plastic and whether or not the material is kept within material circles [23]. As possible solutions closed-loop recycling rates and open-loop recycling rates have been proposed [23], but also quality factors in which the type of material circulation is accounted for [24]. In this study, we will use two Circular Performance Indicators (CPIs) that result from the material flow analysis. These are the net packaging recycling rate and the (average) polymeric purity of the recycled plastics produced. The polymeric purity relates to the applicability of the recycled plastic and, hence, to the type of material cycles the recycled plastics are used in.

This study aims to explore the theoretical limits of a circular recycling value chain for plastic packages in the Netherlands with the currently available technologies and those that are foreseen to be available within the coming five years. These limits are expressed with the two CPIs: net packaging recycling rate and (average) polymer purity of the recycled plastics. This study is based on the previously published material flow analysis of Dutch PC-PPW in 2017, which is extended to also describe the recycling of PI-PPW. In this theoretical study, only technical argumentation is used, whereas economic and social considerations and interrelations are ignored. Legislative aspects are only included in that we consider that recycled plastics need to be as good as contamination-free to be able to return to food contact applications. Furthermore, we propose a classification of recycled plastics to define their applicability. We use this classification as a tool to predict the level of circularity that can be attained. To grasp the full potential of the recycling chain, the combined impact of all improvement options by the stakeholders are studied including synergetic effects.

## 2. Materials and Methods

The Dutch PPW recycling value chain of 2017 was the basis for this study [3]. The model describes the recycling chain of post-consumer PPW (PC-PPW) from packages that are discarded at households to washed milled goods (WMG) as recycling product and generate two CPIs: the net recycling rate and average polymer purity (Appendix A.1). This model was elaborated to include post-industrial plastic packaging waste (PI-PPW) to allow for a comparison with national and European recycling targets (Appendix A.2). Firstly, the general prerequisites for an optimal circular PPW recycling value chain are defined in Section 2.1. Subsequently, these general prerequisites are translated in a description of this

optimal circular recycling system as it could operate with the current recycling technologies and those that will be available to us within 5 years for PC-PPW and PI-PPW in Sections 2.2 and 2.3, respectively. Finally, a classification of the produced recycled plastics is proposed in Section 2.4.

### 2.1. General Prerequisites for an Optimal Circular PPW Recycling Value Chain

We envision that in an optimal circular recycling value chain all stakeholders are completely committed to the performance of this overall system. This system will then have to produce maximum amounts of high-quality recycled plastics that are reused in new packages and related consumer articles, while low-quality side-products and material losses are minimised. Additionally, the overall environmental impacts of the recycling chain need to be minimised and additional environmental burdens, such as unnecessary transport movements, food losses, or emissions of wastewater, should therefore be avoided.

Individual packages will have to be easy to handle in all different stages of the PPW recycling system, which relates to four design aspects. Firstly, all packages in the system should be made of a restricted amount of plastics (two or three polymer types) that can be efficiently separated and processed with simple technologies. This limitation is crucial, to generate sufficiently pure recycled plastics with effective sorting and recycling technologies. Secondly, the design should facilitate collection. Thirdly, the packages should be easily recognised by automatic sorting machines and its dimensions should enable sorting. Fourthly, the packaging components, which are often made from different polymers and materials, should be easy to separate in an efficient manner in the recycling process. Additionally, the packaging materials should be able to fulfil the packaging functions to prevent environmental burdens such as food losses [25]. For the fulfilment of packaging functions, typical essential material properties are strength, stiffness, mechanical buffering, gas and water vapour barriers and temperature resistance. On top of that, the obvious requirement that all packages need to comply with the essential requirements and procurement specifications, which is unfortunately not always the case [26,27].

The collection system should ideally retrieve all the targeted packaging objects and a minimum of non-targeted objects and product residues. The presence of contaminations in the collected material needs to be limited to maintain the quality of the recycled material and avoid losses. Furthermore, the environmental performance of the recycling chain will increase, as contamination causes additional transports, waste streams and emissions.

In an optimal circular recycling value chain, the sorting process needs to maximise the production of mono-material sorted products and minimise mixed plastics. Purer recycled plastic products can be obtained by using multiple complementary separation techniques in the overall sorting and recycling process. However, this generally lowers the yield and often generates lower quality by-products. Hence, these additional separation techniques are only beneficial to the whole circular recycling system when the by-products are recycled as well and no additional material losses occur.

### 2.2. Description of an Optimal Circular PC-PPW Recycling Value Chain

To describe an optimal circular PPW recycling value chain for the Netherlands, the best available technologies were considered as well as those that are likely to be implemented within 5 years. The PPW recycling chain can be improved at the four main stages: packaging producers (packaging design), waste collection companies (collection rate), sorting facilities (technologies, settings, and operations), and the mechanical recycling facilities (technologies, settings, and operations). The proposed ideal circular PPW value chain for all four stages will be described separately.

#### 2.2.1. Design for Recycling

To establish the most ideal set of plastic packaging designs to fit in a circular recycling value chain, two main design aspects have to be considered: performance parameters and recycling requirements. The applied polymers need to be defined to fulfil the packaging functions and

therefore provide the needed water vapour and gas barrier properties, mechanical properties, optical properties, etc. The recycling requirements imply that the polymers should separate easily with the two mainly used separation technologies: near-infrared (NIR) sorting and sink-float separation. Three polymers (PE, PP, PET) were chosen, which encompass different groups of various sub-types and grades, such as the density-based subcategories of architecturally different (level of branching) PEs. Besides, most packages also contain minor components, which in some cases have to be made from other polymers (inks, glues, barrier layers, etc.). For the sake of this idealised study, these components are considered to be negligible.

To model with an ideal set of packaging designs, existing packaging types that were not made of the selected polymers (PET, PE, and PP) were eliminated and replaced with packages made of those selected polymers in three steps.

Firstly, the eliminated packaging types were replaced by alternative packaging types that would qualify the design-for-recycling guidelines and would also fulfil the performance parameters needed for the packaged products (Table A1 in Appendix A). Secondly, black plastic packages were replaced by packages in NIR recognisable colours. Thirdly, packaging types that had a clear environmental benefit and could not all be replaced by suitable alternatives, such as laminated flexibles, were partly replaced by realistic alternatives and partly retained. The designs of the remaining packaging types were improved in the model by adjusting the minor components, such as caps and labels. These components were modelled to be made of PET, PE, and PP only, according to their technical requirements (Table A2). The material composition per packaging type was adjusted accordingly, using the known weights of caps, labels and other packaging components [3,28]. All the design changes are explained in Appendix A.3.

### 2.2.2. Net Collection Rate

The net collection rate of PC-PPW was maximised in the model. Optimal performing Dutch municipalities with full participation rates achieve net collection rates of around 70%, which relates to the maximum apparent selection rate of 70% for participating civilians [16]. The increased collection rate was modelled proportionately per packaging type. The co-collection of other targeted materials (beverage cartons and metals) was proportionately increased with the amount of PPW.

The collection system should ideally retrieve all the targeted packaging objects and a minimum of non-targeted objects and product residues. In reality, however, almost all collection systems yield only a share of the targeted packages present at the households and various contaminants (other materials, attached dirt and contained product residues). The attached moisture and dirt was therefore proportionately increased with the amount of PPW collected. It is expected that the amount of non-packaging plastics and residual waste will not increase proportionally with the amount of collected PPW. In an optimal recycling system, collection services will reduce these non-targeted contributions by performing more quality controls. Therefore, the co-collected amounts of non-packaging plastics and residual waste have only increased to half the amount of the increase in collected PPW. The used equations are provided in Appendix A.4.

Next to separate collection of PPW, part of the Dutch PPW was retrieved via mechanical recovery from MSW. For urban municipalities with a high share of high rise buildings, separate collection systems typically yield low collection rates and high impurity rates. For these municipalities, mechanical recovery from MSW rendered more PPW with fewer contaminations. In the model, the amount of PPW in the MSW that enters recovery operations was not decreased, as it was expected that this MSW will be collected from municipalities without a separate collection system. The amount of MSW that will enter recovery operations was kept the same as the amount in 2017. The amount of PPW in the overall MSW decreased due to the increased separate collection of the PPW.

### 2.2.3. Improved Sorting Process

The sorting process was improved in the model by increasing the sorting fates of the individual packaging types to the correct sorted product and by adding a sorting process to the flexible packaging flow. The sorting fates of all packaging types were raised to the maximal technical feasible level for mono-material sorted products. The residual amounts were redistributed over the remaining sorted products in the same ratio as modelled for 2017 for each packaging type. The applied sorting fates per packaging type are further substantiated in Appendix A.5.1 (Tables A3 and A4).

In the model of the optimal value chain, the flexible packages were further sorted with additional sorting machines to a PE flexible packaging product. This extra sorting step was modelled as an additional sorting step after the conventional sorting process, with the use of specific sorting fates (Table A5). The sorting fates are described and explained in Appendix A.5.2. Moreover, a by-product was formed that consists of the other films and materials that were present in the Film sorted product. This by-product could be added to the Mix sorted product, or the PP flexibles could be even further sorted into a separate sorted product. Both options are modelled. The option with the best results in terms of quality and quantity of the washed milled goods was used to calculate the results of this study. The result of the other option is provided in Appendix B.3.

### 2.2.4. Improved Recycling Processes

The recycling of PET trays was not incorporated in the 2017 model, as these packages were not recycled at that time. The recycling of PET trays is challenging since it is a heterogeneous group of packages that are not designed for recycling, and of which a large sub-group contains multiple polymers (PE, PA, EVOH) that cannot be separated by conventional recycling technologies [29,30]. In a circular PPW recycling value chain, the PET trays should be recyclable, as they are made of mono-PET and designed for recycling (Appendix A.3). The recycling of PET trays was modelled by using the same approach and transfer coefficients for the basic mechanical recycling process as was used for the other sorted products in the model.

### 2.3. Description of an Optimal Circular PI-PPW Recycling Value Chain

Three types of PI-PPW are present in the Netherlands: PET bottles in the deposit-refund system, business to business (B2B) PPW, and plastic packages discarded at companies, offices, institutions, and other out-of-home locations.

The current deposit-refund system (DRS) for large PET bottles (>0.5 L) already performs optimally. Its collection rate is estimated to be 95% and the polymeric purity of the WMG is above 99% [26]. Therefore no realistic improvements are foreseen for this sub-system. Nevertheless, a policy change has been announced to add the small PET bottles (≤0.5 L) to the DRS. This will result in a shift of these small PET bottles from the separate collection and mechanical recovery systems to the DRS. This shift was modelled separately, see Appendix A.6.

The B2B PPW relates to large homogeneous flows of PE film, PP crates, etc. Its sorting and recycling are considered a profitable business activity (Appendix A.2). Therefore, it is assumed that this part of the PI-PPW recycling is already performing near-optimally, and no additional improvements are proposed to this sub-system to create an ideal circular PPW recycling value chain. Due to insufficient data, the average polymer purity cannot be calculated in detail for this sub-system. However, the nature of this material suggests that this material is very pure, and the polymer purity is therefore estimated to be 99%.

The 'other PI-PPW' is a heterogeneous, mixed PPW similar to PC-PPW. It is currently not recycled. In the ideal circular PPW recycling value chain, these packages are collected and recycled as well and treated similarly to PC-PPW. A detailed description of the modelling method is provided in Appendix A.7. Due to insufficient data, the average polymer purity cannot be calculated in detail for this sub-system. However, the nature of this material is expected to be similar to PC-PPW.

Hence we estimated that the average polymer purity of these materials is the same as the average polymer purity of the PC-PPW sub-system.

## *2.4. Application Areas of Recycled Plastics and Corresponding Material Requirements*

A classification of the potential application for the different recycled plastics streams is proposed in Table 1. This classification works with the boundaries of the idealised system elaborated in this manuscript, as well as some current-day realities.

The classification differentiates between food and non-food end applications and is based on expected degradation and contamination of the recycled plastics. Under degradation, we mainly consider the shortening of the polymer chain due to thermomechanical loading, which will result in lower molecular weights and either reduced intrinsic viscosity (IV) for PET or increasing melt flow index (MFI) for sorted PE or sorted PP, which are jointly referred to as polyolefins (PO) in the Table when subject to similar constraints. The nomen mixed polyolefins (MPO) is used for a blend of both PE and PP. Under contamination, we differentiate between polymers, other than the target polymer and non-polymeric contaminants like paper, minerals, and metals. As a further aspect of purity, we include the maximum filter size for the melt filtration step of regranulation.

Furthermore, as we are discussing potential, the classification is a technical one which does not take legislative aspects such as Food Contact Material (FCM) legislation into account. Likewise, the presence of non-intentionally added substances and odours are considered outside the scope of this classification.

More elaboration on the rationale behind the classification, the relation to different conversion processes, and the meaning of concepts like IV and MFI can be found in Appendix A.8.

## *2.5. Boundaries of the Current Study*

The following aspects define the boundaries of the current study:

- The presence of non-intentionally added substances (NIAS) and odour is not considered, the focus is on technical qualities;
- The presence of unavoidable adhesives or barriers, as well as printing inks, are not considered;
- Current (or imminent) state of recycling technology is assumed. For example, the fact that black plastics are not NIR-sortable will not necessarily remain the case;
- Economical aspects are not considered; legal aspects only up to the point that we set 'no contamination' as a condition;

While all of these represent relevant aspects to the reality of plastics recycling, including them would have gone too far for this study, which intends to model the achievable recycling rates and qualities in an optimal circular PPW recycling value chain. Likewise, the authors are fully aware that a similar exercise might be made for a limitation to four polymers types instead of three, for example including polystyrene. However, this is not considered to be the essence of the study.

**Table 1.** Classification of recycled plastics in relation to their applicability.

| Application Type | EoL Fate | Product Types | Typical Acceptable Degradation | Typical Acceptable Contamination |
|---|---|---|---|---|
| Food no contamination (F-NC) | Circular Closed-loop | Bottle-to-bottle (PET, HDPE) Bottle-to-tray (PET) Clear Film-to-film (LDPE) | Very limited PET bottle: IV > 0.76 HDPE bottle: MFI < 3 PET tray: IV > 0.70 LDPE film: 1 < MFI < 6 HDPE film: MFI < 0.4 | Very limited Other polymers: In PET < 50 ppm In PO: Other PO < 1% Non-PO < 50 ppm Non polymers < 50 ppm Specific for film: only clear Melt filtration < 50 μm |
| Non-food Low contamination (NF-LC) | Circular Semi-closed-loop | Bottle-to-bottle (HDPE, PP) Bottle-to-fibre (PET) Non-clear Film-to-film (LDPE, HDPE) —e.g., garbage bags, agricultural film Thin-walled injection moulding products (PP, PE) Pipe (PP) | Limited for PET fibre: IV > 0.62 LDPE, PP film: MFI < 0.4 HDPE, PP bottle: MFI < 3 PP pipe: MFI ≈ 2 Significant for PE, PP injection moulding (MFI can be > 3, up to 30) | Limited PET fibre and LDPE,PP film as F-NC Injection moulding and bottle (PO): Other polymers: Other PO < 5% Non-PO < 1% Non-polymers < 50 ppm Specific for film: all colours Melt filtration < 200 μm |
| Non-food Significant contamination (NF-SC) | Circular Open-loop | Extrusion of bulky products like decking, panels and street furniture (MPO) | Significant MPO: 2 < MFI < 7 | Significant Other polymers (PET, others) < 10–20% (depending on processing conditions) Non polymers < 5% (depending on size) Melt filtration < 800 μm |
| Non-recycling High contamination (NR-HC) | Linear | High-caloric combustibles (cement industry) Incineration with energy recovery | Unlimited | Quasi-unlimited Non-polymer contaminations will affect efficiency of incineration |

## 3. Results

### 3.1. Circular Performance Indicators

The 2017 model of the PPW recycling value chain was elaborated with the recycling of post-industrial plastic packaging waste, and the by-products of PET recycling were included in the calculation of the CPIs. The net packaging recycling rate of this updated PPW recycling chain was 38% and the average polymer purity of the washed milled goods was 93%. These overall CPIs and the contributions are listed in Table 2. The corresponding data is provided in Appendix B.1 (Tables A6 and A7).

**Table 2.** Circular performance indicators of Dutch plastic packaging waste (PPW) recycling value chain in 2017 compared to an optimal circular recycling value chain, [%].

| Circular Performance Indicators | 2017 | Circularity Potential |
|---|---|---|
| PC-PPW net packaging recycling rate | 26 | 69 |
| PI-PPW net packaging recycling rate | 63 | 78 |
| Total PPW net packaging recycling rate | 38 | 72 |
| PC-PPW average polymer purity | 91 | 96 |
| PI-PPW average polymer purity | 97 | 97 |
| Average polymer purity of all washed milled goods from PPW | 93 | 96 |

In the idealised circular recycling chain, a maximum net packaging recycling rate of 72% can be reached, but it should be stressed that this will require drastic measures to be taken by incumbents in a well-concerted action. The produced washed milled goods will have an average polymer purity of 97% (Table 2). These CPIs are the theoretical limits for a circular economy of plastic packages that rely on the full commitment and co-operation of all stakeholders and is based on the currently available technologies (circularity potential).

A limited sensitivity analysis of the model was performed, see Appendix B.4, which revealed that the net collection rate is the parameter that influences the net recycling rate the most (Tables A13 and A14). In case the net collection rate would increase due to improved separation behaviour of the civilians from 70% to 80%, then the total net packaging recycling rate would increase from 72% to 78%.

### 3.2. Amount and Applicability of Recycled Plastics

The amount, polymeric purity, and applicability of the recycled plastics generated by the optimal recycling value chain are listed in Table 3 and can be compared with the same data for the PPW recycling value chain in 2017 (Table A6 in Appendix B). Much more recycled plastics are produced in an optimal circular PPW recycling system as compared to the system in 2017. Furthermore, the average polymer purity is also substantially higher in an optimal recycling system as compared to the system in 2017; 96% compared to 93%, respectively (Table 2).

The rise in polymeric purity can be observed for almost all types of recycled plastics, and this causes sharp increases in the amounts of the highest qualities of recycled plastics, see Figure 1. The category F-NC increases from 32 in 2017 to 93 kton for the optimal recycling chain and the category NF-LC increases from 80 in 2017 to 205 kton for the optimal recycling chain. Conversely, the amount of mediocre quality recycled plastics from the category NF-SC remains almost constant: 81 kton in 2017 and 78 kton in the optimal recycling chain. Hence, within such an optimal recycling system, there will be substantially higher amounts of rPET available for food packaging applications and large amounts of rPE, rPP, and rLDPE available for non-food packaging and related consumer articles.

**Table 3.** The amounts of recycled plastics produced, their polymeric purity and classification of their applicability in an optimal circular PPW recycling value chain (* = estimated).

| Type of PPW | Amount of WMG [Gg] | Polymeric Purity of WMG [%] | Applicability Classification [F-NC, NF-LC, NF-SC, NF-HC] |
|---|---|---|---|
| PC PET bottles | 23 | 99.7 | F-NC |
| PC PE rigid | 26 | 98.1 | NF-LC |
| PC PP rigid | 49 | 97.7 | NF-LC |
| PC PE film | 50 | 98.7 | NF-LC |
| PC Mix (PO mix) | 32 | 93.4 | NF-SC |
| PC PET trays | 51 | 99.8 | F-NC |
| PC PET bottles by-product (PO mix) | 3 | 92.3 | NF-SC |
| PC film by-product (scenario 2) (PO mix) | 16 | 90.9 | NF-SC |
| PI PET bottles deposit-refund (DR) | 19 | 99.9 | F-NC |
| PI PET bottles DR by-product (PO mix) | 2 | 90.7 | NF-SC |
| PI-PPW B2B | 80 | 99 * | NF-LC |
| Other PI (B2B, offices, public space, etc.) | 25 | 97 * | NF-SC |

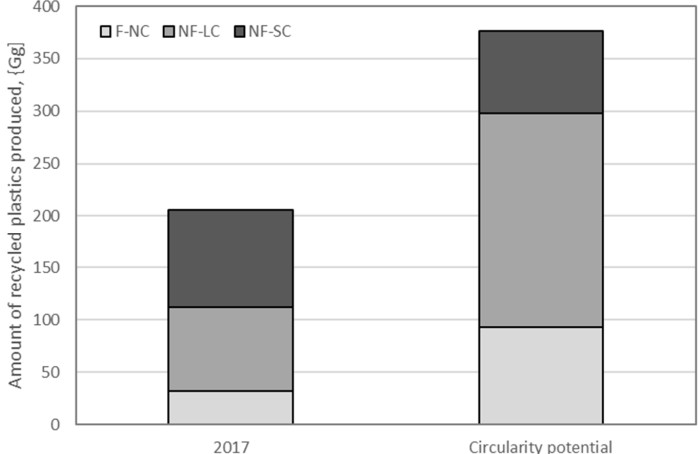

**Figure 1.** The amounts of recycled plastics produced, classified with respect to their applicability in 2017 and in the optimal recycling value chain.

## 4. Discussion

### 4.1. Towards a More Circular Dutch PPW Recycling Value Chain

This study shows that although the Dutch recycling system for PPW is currently one of the best performing systems globally, it can still be improved substantially with existing technologies. Both CPIs can be improved: the net packaging recycling rate can be improved from 37 to 72% and the average polymer purity can be improved from 93 to 97%. The recycling targets for plastic packages have been set to 50% by the end of 2025 and 55% by the end of 2030 [31]. Hence, it would appear that the European recycling targets can easily be achieved. However, in order to achieve the optimal circular economy for plastic packages, drastic improvement measures have to be implemented by all stakeholders. Not only would this signify major investments in new packaging machines at the producers, new sorting equipment at sorting facilities, and new recycling equipment at recycling facilities, these investments also need to be well-orchestrated. Since the benefits of these investments are limited for the individual stakeholders, these investments will not be made, without clear regulations and control mechanisms.

The net packaging recycling rate is strongly influenced by the net collection rate. Autonomous improvements in the collection systems by municipalities can improve the net packaging recycling rate to only 45% for PC-PPW (Appendix B.2), whereas the net packaging recycling rate can be raised further to 69% for PC-PPW if the other stakeholders also take the appropriate concerted actions. Thus, well-aligned improvement measures taken by all the stakeholders result in a synergistically better recycling rate compared to the recycling rate that can be achieved when only municipalities optimise

the collection. The average polymer purity is mainly affected by the design of the packages, and to a lesser extent by the sorting and recycling processes. Autonomous design-for-recycling measures taken by the producers can increase the average polymer purity of the PC-PPW recycling chain from 91% to 94% (Appendix B.2, Table A9), whereas with combined and well-aligned measures of all the stakeholders, an average polymer purity of 97% can be achieved. Hence, for both CPIs synergy can be obtained when the stakeholders take coordinated improvement measures. Concerted action of all stakeholders simultaneously is important to achieve the recycling targets set by the EU and to progress towards a quantitively and qualitatively high performing circular PPW recycling value chain. Nonetheless, this implies that close coordination and strict governance is required to achieve this ideal plastic packaging recycling value chain.

### 4.2. Recycled Plastic Markets

Traditionally, plastic markets are divided by sector (packaging, building and construction (B&C), automotive, consumer goods, etc.), wherein the packaging sector represents around 40% of the annual turnover of (virgin) plastics. Due to the short shelf-life of most packaging plastics, they represent around 60% of plastic waste. It would, therefore, make sense to want to return PPW as secondary resources to the packaging sector. This sector is, however, dominated by food packaging, where food safety concerns lead to stringent regulations for FCM. Achieving FCM approval is often more a regulatory concern than an actual technical one [8]. In many cases, FCM grade can be achieved from a technical perspective (contamination levels, migration limits), but formal approvals are extremely challenging to obtain. For our idealised system to work, legislation and technical practice would be assumed to be much better aligned. Even so, there exists an important difference between PET and PO when considering the long-term: it is inherent to the PET recycling process that the polymer chains are 'repaired' during recycling, while this is not possible for PP and PE. Therefore, the PO will typically continue to lose chain length over multiple recycling cycles (we disregard here the possibility for HDPE to crosslink due to degradation, which will have its own consequences) and will have more need of a remaining influx of virgin materials to compensate this.

For those plastics unable to be returned to the packaging industry, other markets must be considered. In general, these markets are not unwilling to take up the recycled plastics, but they are bound to the technical specifications for their materials, as outlined in Table 1. Many materials originally used for film, for example, is extremely challenging to re-use injection moulding. Similarly, PET is an excellent barrier material, but typically quite brittle, which gives it little applications outside of packaging, except for fibre. The most promising alternative markets would be building and construction (B&C: pipes, bulky parts, panels, large injection moulding parts), in which products typically use large volumes and food contact is rarely an issue. It might even be argued that using these materials in B&C is a more sustainable destination, as the product lifetimes are typically much longer (up to 30 years).

Since this is a technical study, economic and political aspects are not considered. But these will influence the development of the value chain for PPW in the future. For instance, the growth of the recycling rates can be frustrated by the saturation of markets for lower quality recycled plastics (especially PC-MIX). Additionally, the current low prices for virgin plastics will hamper the further expansion of the recycling industry.

### 4.3. Variations in the Structure of an Ideal Circular Recycling Value Chain

Although most of the choices made in the design of the optimal circular recycling value chain are self-evident, a few of them are more debatable. Four of these variations in the structure of the optimal value chain will be discussed below.

#### 4.3.1. Creating a Separate Sorted Product for PP Flexible Film

Two scenarios for the additional sorting of flexible packages are explored (See Appendix B.3). In the first scenario, a separate sorted PP flexible film product is created. The model predicts that this

renders a PP flexible product with a low polymeric purity (84%) as a consequence of a large amount of PE flexible film products in the feedstock. Additionally, a mixed polyolefin by-product is produced with a relatively low polymer purity of 84% (NF-LC, Table A10), which can be added to the PC-PPW Mix. In the second scenario, the PP flexibles and the laminated flexibles are jointly sorted from the sorted film product to form a mixed polyolefin by-product. This by-product has a polymeric purity of 91% (NF-LC, Table A10) and can also be added to the PC-PPW Mix (Table A11). Therefore, with the current sorting technology and type of packages on the market, it is best to keep the PP flexibles in the mixed plastics.

### 4.3.2. The Continued Need for Mixed Plastics

There is still a need for a mixed plastic sorted product (PC-PPW Mix, NF-LC) in the optimal circular recycling value chain. However, its relative importance is reduced. Only 12% of the mass of the separately collected PC-PPW is sorted to the Mix in the ideal circular recycling system compared to 26% in 2017 (Table A12). The mixed plastics that originate from sorting PC-PPW is mostly composed of flexibles, non-packaging plastics, PET trays and residual waste. The relative large contribution of flexibles to the mixed plastics stems from the mediocre sorting efficiencies for flexibles in wind sifters and ballistic separators.

### 4.3.3. Adding Small PET Bottles to the Deposit Refund System

The addition of the small PET bottles ($\leq$0.5 L) to the DRS results in a shift in the amounts of recycled PET being produced. The amount of recycled PET from PC-PPW reduces from 23 to 16 Gg and the recycled PET from DRS increases from 19 to 29 Gg. This shift doesn't affect the net packaging recycling rate of the overall PPW recycling value chain, but the amount of recycled PET that is suitable for food high demanding applications increases with 10 Gg.

### 4.4. Limits to Circularity

Although the current recycling chain for PPW in the Netherlands is one of the more advanced systems globally, it is still far from an optimal circular economy for PPW. To attain the latter, unprecedented efforts of all stakeholders in a tight orchestration are required, which will signify massive investments of all stakeholders. Nevertheless, even after these unparalleled efforts have been delivered, this maximal achievable recycling system is still highly dependent on fossil feedstock for the production of food-grade packages and on non-food aftermarkets for the application of recycled plastics. Moreover, the maximum achievable recycling rate for plastic packages is just 72%. Only the PET food packages are potentially circularly recyclable into new food-packages. Although, even for this material, precautionary measures have to be taken to avoid the accumulation of contaminants [32–35]. The most widely used food-packaging polymers (PE and PP) can still only be used once in the circular economy based on mechanical recycling technologies, in large part due to legislative restrictions and the current inability to sort food-grade from non-food-grade materials. Most of these PE and PP food packages can also not be re-designed into PET food packages for packaging technological reasons such as temperature resistance, light transmission etc. Furthermore, also the reduction of plastic food packages in our modern society has its limitation and, when uncarefully executed, results in more food waste [36].

Although previous studies have reasoned that such an open-loop recycling system, which relies on fossil oil input and non-food after markets, might be the environmentally favoured option [37,38], both legislators and retail organisations strive towards more closed-loop recycling systems. Demanding and implementing recycled content in plastic packages is regarded by them as the most tangible and convincing measure to reduce the environmental impact of plastic packages. This quest for more circularity, however, entails unprecedented efforts and investments that need to be considered as well.

To progress beyond this 'optimal circular economy value chain', disruptive innovations in design, sorting and recycling are required. Hitherto multiple measures and innovations have been proposed; to use marking technologies to assist sorting [39–41], to use magnetic density separation to replace sorting and recycling for rigid packaging plastics [42], to use new sorting logarithmic sorting technologies for rigid plastic flakes [43], to use fluorescent imaging to sort plastics [44], to chemically recycle flexibles and mixed plastics [45,46], to make black plastics NIR-sortable [44,47], etc. Undoubtedly, more innovations and measures will be proposed in the future. The challenge will be to select and align those innovations to achieve societal targets and balance the interests of the stakeholders.

## 5. Conclusions

We have considered an idealised circular economy for plastics packaging, wherein all plastic packaging is made of PE, PP, or PET and the four main stages of the PPW recycling value chain (design, collection, sorting, and mechanical recycling) cooperate without reservation. The technical limit for the recycling rate of plastic packages is 72% within such an idealised circular economy, using the currently available recycling technologies and those that are foreseen within the coming five years. To achieve this circularity potential, unprecedented offers have to be made by all stakeholders. Nonetheless, this optimal circular economy for plastic packages still relies, to a large extent, on fossil oil as an input for food-grade packaging materials and non-food packaging and non-packaging aftermarkets.

Future research would be welcomed on the acceptability of the required efforts by the stakeholders in relation to the circularity that can be achieved. Additionally, the impact of new disruptive technologies and policies (reusable packages) on the whole value chain could be explored.

**Author Contributions:** Conceptualization, M.T.B. and E.U.T.v.V.; methodology of modelling, M.T.B.; definition of the ultimate design for recycling options, M.T.B. and R.t.K.; definition of the applicability of recycled plastics, K.R.; data analysis, M.T.B.; writing—original draft preparation, M.T.B., E.U.T.v.V. and K.R.; writing—review and editing, E.U.T.v.V., M.T.B., R.t.K. and K.R. All authors have read and agreed to the published version of the manuscript.

**Funding:** This research was funded by Wageningen Food & Biobased Research.

**Conflicts of Interest:** The authors declare no conflict of interest.

## Appendix A. Additional Information Research Method

*Appendix A.1. Equations of Circular Performance Indicators (CPIs)*

Two equations were used to calculate both CPIs, see Equations (A1) and (A2). The net recycling rate for plastic packages ($R_{\text{net Plast.Pack.}}$) was calculated by dividing the net mass of recycled packaging materials (for plastics this is the intermediate product of washed milled goods) ($M_{\text{Rec. Plast. Pack.}}$) by the mass of plastic packages placed on the market) ($M_{\text{Plast. Pack on market}}$·).

$$R_{\text{net Plast.Pack.}} = M_{\text{Rec. Plast. Pack.}}/M_{\text{Plast. Pack on market}} \tag{A1}$$

The average polymer purity (APP) is calculated from the sum of the polymeric purities of the recycled plastics ($PP_{\text{RPi}}$) made divided by the number of recycled plastic products (n) made by the complete recycling network.

$$APP = (PP_{\text{RP1}} + PP_{\text{RP2}} + \dots PP_{\text{RPn}})/n \tag{A2}$$

*Appendix A.2. Model Updates Dutch 2017 PPW Recycling Value Chain*

The Dutch 2017 PPW recycling value chain was extended to allow for a comparison of the calculated net packaging recycling rate with the recycling targets set by the European Commission and Dutch government. These model extensions affected the net packaging recycling rate (Equation (A1)), as the amount of plastic packaging material on the market and the mass of the recycled plastic

packaging material both changed. The average polymer purity (Equation (A2)) was also affected, as recycled products were added to the calculation.

In this new version of the model, all plastic packages on the Dutch market are included. Hence, the officially registered amount of plastic packaging material on the market could be used in contrast with previous versions of the model [3]. The total mass of the plastic packaging material on the market is registered by Nedvang and was 512 kton in 2017 [48].

The recycling targets set by the European Commission and the Dutch government include all recycled packaging plastics. The polyolefin by-products from the recycling of PET bottles are recyclable and therefore included in the mass of recycled plastic packaging material. The updated model also included the masses of the recycled plastic packaging material originating from three types of post-industrial plastic packaging waste (PI-PPW): PET bottles in the deposit-refund system, business to business (B2B) PPW and plastic packages discarded at companies, offices, institutions and other out-of-home locations.

PET bottles from the deposit-refund system are collected, counted, and mechanically recycled. The amount of collected and counted PET bottles from the deposit-refund system is 23 gross Gg, with a moisture and dirt content of $8 \pm 2\%$ and thus a net amount of 21 Gg of PET bottles [49]. The mechanical recycling process is modelled with the same calculations and transfer coefficients for mechanical recycling as in the model of the PC-PPW recycling value chain of 2017, and only the material composition of these specific PET bottles was included to the model as additional data [3]. The used composition of PET bottles in the deposit-refund system was $92 \pm 2\%$ PET, $2 \pm 2\%$ PP, $6 \pm 2\%$ PE and a small amount of paper ($0 \pm 1\%$) [28].

The recycling of B2B PPW is not yet studied in detail, and thus only general data was available. In 2017, the collected amount of other PI-PPW, besides the PET bottles from the deposit-refund system, was 91 Gg [49]. The collection and sorting of homogeneous PI-PPW is typically a profitable business activity and doesn't require subsidies or funds of an extended producer responsibility scheme (EPR), whereas the heterogeneous PI-PPW do need additional funding, but the Dutch EPR scheme operator doesn't provide these. Since, the Dutch EPR scheme operator reports that in the Netherlands, in 2017, 91 kton PI-PPW was collected it is assumed that this was homogeneous B2B PPW material, such as PE pallet wrap film, jerry cans, crates and intermediate bulk containers (IBCs). The precise amounts and material compositions of these packages are unknown, but in order for their recycling to be profitable, they need to be fairly pure. This lack of data resulted in a more general modelling approach, with the estimation that 10 to 15% of the collected material was lost during mechanical recycling. These losses include moisture and dirt and the removal of non-targeted materials. Due to the lack of data, the average polymer purity cannot be calculated in detail for this sub-system. However, the nature of this material suggests that this material is very pure, and the polymer purity is therefore estimated to be 99%. The plastic packages that are discarded at companies, institutions, railway stations, offices, etc., (named 'other PI-PPW') were too heterogeneous for profitable recycling and therefore not collected and recycled in 2017.

*Appendix A.3. Design for Recycling*

The design for recycling measures was rationalised per packaging type. Several packaging types were eliminated altogether and replaced by alternative packaging types, these are listed in Table A1. Packaging types were eliminated in case they hinder the sorting and recycling of targeted plastic packages or in case their main polymer was a non-targeted polymer. The eliminated packaging types were replaced with alternative packaging types that could fulfil all the performance criteria (water vapour & gas permeability, mechanical properties, optical properties, food safety, thermal stability, etc.) and simultaneously fulfil all the requirements for sorting and recycling facilities. The packaging design of the continued packaging types was improved as is explained in Table A2.

**Table A1.** The design changes that relate to eliminated packaging types and the packaging types for which they are replaced, including explanations.

| Eliminated Packaging Type | Replacement | Explanation |
|---|---|---|
| PET bottle coloured ≤ 0.5 litre | PET bottle clear ≤ 0.5 litre | Coloured PET bottles could be replaced by transparent PET bottles. |
| PET bottle coloured > 0.5 litre | PET bottle clear > 0.5 litre | |
| PS beverage bottles | PET bottle clear ≤ 0.5 litre | Mainly small bottles, these could be replaced by transparent PET bottles. |
| PS thermoforms & rigids | PP thermoforms & rigids | Mainly yoghurt pots, creme fraiche pots, whipping cream pots, cookie trays. Based on the form of these packages it would make sense to replace these with PP. |
| PVC thermoforms & rigids | PET thermoforms & rigids | Mainly transparent blister packages. These can best be replaced with PET. |
| PET flexible packages > A4 | PE flexible packages > A4 | PET film packages are fairly uncommon, these packaging types can be replaced by PE film. |
| PET flexible packages < A4 | PE flexible packages < A4 | |
| PS flexible packages < A4 | PP flexible packages < A4 | PS film packages are fairly uncommon, these packaging types can be replaced by PP film for transparency and gloss. |
| PVC flexible packages > A4 | PE flexible packages > A4 | PVC film packages are used for their puncture resistance and transparency. They can be replaced by PE film, but the puncture resistance and transparency will be less. PVC stretch wrap is used for packaging sprouts, perforated PE film can be used for these packages as well. Moreover, these packages might be replaced by alternative packaging concepts. |
| PVC flexible packages < A4 | PE flexible packages < A4 | |
| Rigid packages made from non-NIR identifiable plastics | PET, PE, PP thermoforms & rigids in ratio of original market share | The packages are made of the same material, but are either coloured in a different colour or coloured black with a detectable black colourant. |
| Flexible packages made from non-NIR identifiable plastics > A4 | PE flexible packages > A4 | |
| Flexible packages made from non-NIR identifiable plastics < A4 | PE flexible packages < A4 | |
| Misc. plastics (PC, PLA, etc.) | 50% PE flexible packages < A4 50% PET, PE, PP thermoforms & rigids in ratio of original market share | These materials could be used in rigid and in flexible packaging. The ratio in which they are present is unknown, hence they are equally divided over both categories. |
| Laminated flexible packages and blisters | Partly replaced: 10% PE flexible packages < A4 10% PE thermoforms & rigids 10% other packages concepts, which will be collected via other collection schemes (paper, metal, etc.) and are therefore deleted from the model. 70% of the laminated flexible packages and blisters are considered to be environmentally beneficial or could not be replaced due to legislation. These are kept in the model as laminates. | Some laminated films (such as pouches) can be replaced by simple PE film. Chewing gum blisters etc. could be replaced by PE thermoforms & rigids. Some laminated can be replaced by alternative packaging concepts, e.g., beverage carton, cans, etc. such as soup pouches. Some laminates have a clear added environmental benefits: good product protection (less food waste) and lightweight packages. This should be considered in the choice to replace these packages. Therefore, not all laminates can and should be replaced, unless alternative packaging solutions are developed. Moreover, PVC drug blisters are registered as packaging material for specific drugs and changing them would require a new registration procedure. In the future even more laminated films might be replaced by mono-material films due to other packaging strategies, such as the use of anti-oxidants and shortening the shelf life of products. |
| EPS trays | PET thermoforms & rigids | Mainly meat trays. These packages are already banned in the Netherlands so are no longer common and could be replaced by PET trays. |
| EPS blocks | Deleted from the PPW stream | Likely to be replaced by pressed carton board, folding board or other new materials, and collected via dedicated systems. In case the EPS blocks cannot be replaced by other materials with the same mechanical buffering characteristics, they should be collected via a separate collection system. So deleted from the PPW stream |
| Silicone sealant cartridges (Rigid plastic tubes with silicone paste) | These packages are not replaced, but 100% collected via the municipal solid waste. | These packages are considered contaminants due to their product content. Therefore, sorting facilities will make sure that these packages are not present in the sorted products by taking them out manually. For modelling purposes, these packages are therefore collected via the municipal solid waste and not recovered. |

**Table A2.** Packaging composition of the continued packaging types, including an explanation of the chosen composition [1,2].

| Continued Packaging Type | Material Composition | Explanation |
|---|---|---|
| PET bottle clear ≤ 0.5 litre | 85% PET<br>2% PP<br>13% PE | PET body with a PE cap and a PP label. This material choice will result in mono-material by-products: the wind-sifted fraction will consist of PP and the sink-float separated fraction will consist of PE. Material ratios are based on average packaging designs. |
| PET bottle clear > 0.5 litre | 91% PET<br>2% PP<br>7% PE | |
| PE beverage bottles | 2% PP<br>98% PE | PE body, with a PP label and a PE cap. |
| PP beverage bottles | 85% PP<br>15% PE | PP body with a PP label and a PE cap. [3] |
| PET non-beverage bottles | 83% PET<br>2% PP<br>15% PE | Same as PET bottle clear > 0.5 litre, with a little heavier cap based on average packaging designs. |
| PE non-beverage bottles | 2% PP<br>98% PE | Same as PE beverage bottles. |
| PP non-beverage bottles | 85% PP<br>15% PE | Same as PP beverage bottles. |
| PET thermoforms & rigids | 100% PET | Mono-PET tray with PET based top-lid. If a sealing agent is needed, this should be water soluble in the mechanical recycling process. [4] |
| PE thermoforms & rigids | 2% PP<br>98% PE | PE body with only a PP label. |
| PP thermoforms & rigids | 100% PP | PP body with PP in mould label |
| Carriage bags (PE) > A4 | 100% PE film | No adhesive labels, glued on labels, etc. Only film material, with prints directly on the film. |
| Carriage bags (PE) < A4 | 100% PE film | |
| PE flexible packages > A4 | 100% PE film | |
| PE flexible packages < A4 | 100% PE film | |
| PP flexible packages > A4 | 100% PP film | |
| PP flexible packages < A4 | 100% PP film | |
| Laminated flexible packages and blisters | Not changed. | Same as in original model [3]. |

[1] This table is focused on the material composition of the packaging components. There are more design for recycling guidelines that can be followed. An example is the use of sleeves and large labels that should be avoided as they could hamper the material recognition in (near-infrared) NIR sorting. [2] The use of inks, glues and other minor packaging components are outside the scope of the model, and therefore not defined in the table. However, in design for recycling these should be considered as well. It is advised to use inks that do not bleed, and that are not toxic or hazardous. [3] The PP bottles were modelled with PP labels and PE caps. It would also be possible to use PP caps. In that case, the whole bottle would be made of one polymer type. However, these PP materials would have to be made of different grades with different tacticity and MFI values. Mixing grades will result in a more average PP recycled product [50]. We choose to model the caps to be made of PE. [4] There are several types of mono-PET trays on the market. The clamshells and top-sealed trays for fruits, vegetables and nuts are true mono-A-PET systems. However the current so-called "mono-PET meat trays" have a PE sealing layer on the flange [30]. This layer is needed to seal the top-lid in a reliable and fast manner on the tray. Hence, the current mono-PET meat trays introduce a small amount of PE to the PET materials, which results in hazy light grey rPET. Furthermore, the applied top-films are either composed of PET-PE, PET-PA-PE or PET-EVOH-PE and a hence also a source of polymeric contaminants [30]. In an optimal circular PPW recycling value chain, all PET trays are first mechanically recycled and subsequently de-polymerised, purified and re-polymerised to obtain food-grade PET resins.

*Appendix A.4. Equations to Calculate the Separately Collected Amount of Materials*

The amount of collected packages per packaging type ($_{Mcoll.pack.type}$) is calculated by multiplying the discarded amount of these packages by the households ($_{Mdisc.pack.type}$) with the collection fate of these individual packaging types in a circular PPW recycling value chain ($CF_{pack.type.circular}$), see Equation (A3). The collection fate of individual packaging types in a circular PPW recycling value chain is calculated using Equation (A4), with a net collection rate in a circular PPW recycling value

chain (CR$_{net\ circular}$) of 70% multiplied with the collection fate of that individual packaging type in 2017 (CF$_{pack\ type\ 2017}$) and divided by the average collection fate in 2017 (CF$_{average\ 2017}$).

$$M_{coll.pack.type} = M_{disc.pack.type} \times CF_{pack.type.circular} \tag{A3}$$

$$CF_{pack.type.circular} = CR_{net\ circular} \times CF_{pack.type.2017}/CF_{average\ 2017} \tag{A4}$$

The amount of co-collected materials was modelled proportionally to the amount of plastic packaging waste, as explained in the paper. As an example, the calculation method of the co-collected amount of beverage cartons (M$_{BC.circular}$) is shown in Equation (A5). It is calculated by multiplying the amount of co-collected beverage cartons in 2017 (M$_{BC\ 2017}$) with the amount of collected plastic packaging waste in a fully circular value chain (M$_{PPW\ circular}$) divided by the amount of plastic packaging waste in 2017 (M$_{PPW\ 2017}$). The co-collected amounts of non-packaging plastics and residual waste have only been increased halve compared to the increase in collected PPW as explained in Section 2.2.2.

$$M_{BC.circular} = M_{BC\ 2017} \times M_{PPW.circular}/M_{PPW\ 2017} \tag{A5}$$

*Appendix A.5. Sorting Process*

Appendix A.5.1. Maximal Technical Feasible Sorting Fates

The sorting fates of all plastic packaging types were adjusted to the maximal technical feasible amounts in the mono-material sorted products (PET bottles, PE, PP), see Table A3. It was estimated that the near-infrared (NIR) sorting technology has a maximum efficiency of 90%. Furthermore, the highest previous recorded sorting fate for a rigid plastic package was 91% for small PET bottles [51] (Table L). Hence, the maximum feasible sorting fate for PET, PE and PP rigid packages were estimated to be 90%. The sorting fates of the flexible packaging are based on the efficiency of the wind sifters and ballistic sorters. We estimated the maximum feasible sorting fate to be 80% for flexible packages larger than A4 and to be 50% for flexible packages smaller than A4. Hitherto the highest recorded sorting fate for flexible packages was 58% for PE carriage bags [3,51] (Table L) and improved technologies should be able to increase this sorting fate to 80%. The sorting fate of laminated film packages was also estimated to be 50%, as these packages are mainly smaller than A4.

**Table A3.** Sorting fates of the packaging types to the targeted sorted products in an optimal PPW recycling value chain (only relates to the sorting of separately collected PPW).

| Packaging Type | Targeted Sorted Product | Sorting Fate to Targeted Sorted Product [%] |
|---|---|---|
| PET bottle clear ≤ 0.5 litre | PET bottles | 90 |
| PET bottle clear > 0.5 litre | PET bottles | 90 |
| PE beverage bottles | PE rigids | 90 |
| PP beverage bottles | PP rigids | 90 |
| PET non-beverage bottles | PET bottles | 90 |
| PE non-beverage bottles | PE rigids | 90 |
| PP non-beverage bottles | PP rigids | 90 |
| PET thermoforms & rigids | PET trays | 90 |
| PE thermoforms & rigids | PE rigids | 90 |
| PP thermoforms & rigids | PP rigids | 90 |
| Carriage bags (PE) > A4 | Film | 80 |
| Carriage bags (PE) < A4 | Film | 50 |
| PE flexible packages > A4 | Film | 80 |
| PE flexible packages < A4 | Film | 50 |
| PP flexible packages > A4 | Film | 80 |
| PP flexible packages < A4 | Film | 50 |
| Laminated flexible packages and blisters | Film | 50 |

The recovery and sorting process of PPW in MSW was also improved by increasing the sorting fates of the individual packages (Table A4). The recovery and sorting of PPW from MSW is a two-step process that is modelled with one sorting fate. The sorting fate is, therefore, lower than the sorting fate

for the sorting of separately collected packages. The sorting fate of PET, PE and PP rigid packages estimated to be 70%. The sorting fate of PE and PP film packages estimated to be 45% for both packages >A4 and <A4. The sorting fate of laminated film packages was also estimated to be 45%. Again the previous maximum recorded sorting fates [51] (Table M) were used substantiate the selected values.

**Table A4.** Sorting fates of the packaging types to the targeted sorted products in an optimal PPW recycling value chain (these fates relate to both the mechanical recovery from MSW and the subsequent sorting).

| Packaging Type | Targeted Sorted Product | Sorting Fate to Targeted Sorted Product [%] |
|---|---|---|
| PET bottle clear ≤ 0.5 litre | PET bottles | 70 |
| PET bottle clear > 0.5 litre | PET bottles | 70 |
| PE beverage bottles | PE rigids | 70 |
| PP beverage bottles | PP rigids | 70 |
| PET non-beverage bottles | PET bottles | 70 |
| PE non-beverage bottles | PE rigids | 70 |
| PP non-beverage bottles | PP rigids | 70 |
| PET thermoforms & rigids | PET trays | 70 |
| PE thermoforms & rigids | PE rigids | 70 |
| PP thermoforms & rigids | PP rigids | 70 |
| Carriage bags (PE) > A4 | Film | 45 |
| Carriage bags (PE) < A4 | Film | 45 |
| PE flexible packages > A4 | Film | 45 |
| PE flexible packages < A4 | Film | 45 |
| PP flexible packages > A4 | Film | 45 |
| PP flexible packages < A4 | Film | 45 |
| Laminated flexible packages and blisters | Film | 45 |

Appendix A.5.2. Sorting Fates of the Additional Film Sorting Process

The additional sorting process for flexible packages is performed with NIR-sorting technology. Hence, the sorting fates were estimated based on the maximal technical feasible efficiencies of this technology (Table A5). The maximal efficiency of this technology is estimated to be 90% for the targeted materials. Therefore, the sorting fate of PE flexible objects to the PE sorted product and of the PP flexible objects to the PP sorted products was estimated to be 90%. The sorting fates of non-targeted materials towards the PE-film and PP-film sorted products were estimated to be 2%. The remaining material was added to the Mix sorted product.

**Table A5.** Sorting fates of the additional sorting process for flexible packages in an optimal PPW recycling value chain.

| Packaging Type | Sorting Fate to PE Film [%] | Sorting Fate to PP Film [%] | Sorting Fate to Mix [%] |
|---|---|---|---|
| PET bottle clear ≤ 0.5 litre | 2 | 2 | 96 |
| PET bottle clear > 0.5 litre | 2 | 2 | 96 |
| PE beverage bottles | 90 | 2 | 8 |
| PP beverage bottles | 2 | 90 | 8 |
| PET non-beverage bottles | 2 | 2 | 96 |
| PE non-beverage bottles | 90 | 2 | 8 |
| PP non-beverage bottles | 2 | 90 | 8 |
| PET thermoforms & rigids | 2 | 2 | 96 |
| PE thermoforms & rigids | 90 | 2 | 8 |
| PP thermoforms & rigids | 2 | 90 | 8 |
| Carriage bags (PE) > A4 | 90 | 2 | 8 |
| Carriage bags (PE) < A4 | 90 | 2 | 8 |
| PE flexible packages > A4 | 90 | 2 | 8 |
| PE flexible packages < A4 | 90 | 2 | 8 |
| PP flexible packages > A4 | 2 | 90 | 8 |
| PP flexible packages < A4 | 2 | 90 | 8 |
| Laminated flexible packages and blisters | 25 | 25 | 75 |
| PET non-packages | 2 | 2 | 96 |
| PE rigid non-packages | 90 | 2 | 8 |
| PE film non-packages | 90 | 2 | 8 |

**Table A5.** *Cont.*

| Packaging Type | Sorting Fate to PE Film [%] | Sorting Fate to PP Film [%] | Sorting Fate to Mix [%] |
|---|---|---|---|
| PP non-packages | 2 | 90 | 8 |
| PVC non-packages | 2 | 2 | 96 |
| PS non-packages | 2 | 2 | 96 |
| non-NIR identifiable non-packages | 2 | 2 | 96 |
| Beverage cartons | 2 | 2 | 96 |
| Metals | 2 | 2 | 96 |
| Organics & undefined | 2 | 2 | 96 |
| Textiles | 2 | 2 | 96 |
| Paper & cardboard | 2 | 2 | 96 |
| Glass | 2 | 2 | 96 |

Two scenarios were calculated, as described in the paper:

- Scenario 1. Three sorted products are produced: PE film, PP film and Mix.
- Scenario 2. Two sorted products are produced: PE film and Mix (which included the PP flexible packages).

*Appendix A.6. Deposit Refund on Small PET Bottles*

In this study, the inclusion of small (≤0.5 L) PET bottles in the deposit-refund system was modelled as a separate scenario. Not all PET bottles are included in the deposit-refund system, for instance, juice bottles are excluded from this system. We assumed that the same percentage of PET bottles ≤0.5 L were added to the deposit-refund system as the PET bottles >0.5 L in 2017. This percentage was estimated to be 80%, which was based on the division in 2017 of about 80% PET bottles >0.5 L in the deposit-refund system and 20% in the PC-PWW recycling system.

*Appendix A.7. Other PI-PPW*

The amount of 'other PI-PPW' was estimated to be 60 net kton, which equals the difference between the total amount of plastic packaging material on the market, and the amounts of plastic packaging materials that are collected via the other collection routes (PC-PPW, deposit-refund system and B2B PI-PPW). The collection yield of the 'other PI-PPW' was estimated to be 50%. The sorting process was estimated to have the same efficiency as the sorting process of PC-PPW, which was 97% for the plastic packaging types to the plastic sorted products (PET bottles, PE, PP, PET trays, Film and Mix). The mechanical recycling yield was also estimated to be same as for PC-PPW (88%). The mechanical recycling yield was calculated by dividing the amount of produced washed milled goods by the net amount of plastic packaging in the sorted products.

*Appendix A.8. Rationale of the Classification*

Making a neat classification for recycled plastics is extremely challenging, as requirements for the polymers are often very specific to a given product or even a company-specific execution of it. Nonetheless, we have attempted a more general classification. Terminology implying an assessment of value is purposefully avoided; the open-loop application does not automatically imply 'downcycling' and even the bulky applications like street furniture have their sustainable merits, as these are very long-lasting products, often having a product life of several decades, whereas the packaging they originate from has a shelf life of months.

The classification made is based on technical requirements and quality of the recycled goods and does not take legislative aspects into account, as these are subject to rapid evolution. Currently, food-grade recycling is almost exclusively authorized in the EU for bottle PET, but it is expected that similar authorizations will follow for PP, HDPE and LDPE [8].

As input streams to the classification, we have not considered PET trays, as they are currently not recycled. Likewise, we have not considered multilayer products, as all current sorting systems send these to the residue (for incineration or landfill) [52]. All of this may change in future scenario's.

Appendix A.8.1. The Different Polymer Processing Options

Different types of polymer products are manufactured through different processes, which in turn have different requirements in terms of polymer flow.

For PO this is industrially summarized by MFI (g/10 min), a simply measured property that is inverse to the viscosity (= the resistance to flow): low MFI values typically mean high viscosity, high melt strength and low flow; High MFI values mean low viscosity, low melt strength and high flow [53,54]. Injection moulding requires high flows, to quickly fill all cavities in the mould. Extrusion blow moulding (of HDPE or PP bottles) requires an average MFI, as the polymer must flow well enough to be blown up against the mould interior, but must also have sufficient melt strength to keep structural integrity during the melt-based forming step. The latter is also valid for sheet or pipe extrusion, which requires low MFI values as the extruded melt must support its own weight for a brief while. Materials for film blowing are in between extrusion and extrusion blow moulding: they must be stretched thinner than in bottle production, but they still need to support the integrity of the blown bubble. The bulky products listed for NF-SC are manufactured either through extrusion or the related technique of intrusion. These are slow processes, which make them somewhat forgiving towards the upper limit for MFI values [24].

PET bottles are first injection-moulded as a pre-form and then stretch blow moulded into the final form. PET trays are first extruded as sheets and then thermoformed. In PET qualities, a classification based on intrinsic viscosity (IV) is generally used, rather than MFI. IV is a measure for how long it takes the dissolved polymer to dilute through a capillary, compared to the pure solvent. As such, IV is a dimensionless value and equivalent to the molecular weight of the polymer. High IV values represent a longer polymer chain and as such a higher quality PET, which is more expensive to manufacture, as it takes more time in the post-condensation stage.

Typical values for MFI and IV limits are given in Table 1; they are based on literature and professional exchanges with industry. MFI is characterized by a temperature and weight of testing. The values used in Table 1 are the ones at 190 °C, 5 kg for PE and MPO and at 230 °C, 5kg for the higher-melting PP. The recorded MFI ranges will appear higher for LDPE than HDPE. This is because the measurement temperature is further above the melt temperature for LDPE, causing it to flow better at the same temperature. The 190 °C is the universal test temperature, this does not impose such temperatures on the process itself, even if LDPE's inherent strain hardening would allow it. Indeed, LDPE film can be manufactured at lower temperatures.

Do be aware that these are target values which are used as rules of thumb. With adapting processing conditions such as temperatures and pressures, it is definitely possible to also process materials outside of these specifications.

Appendix A.8.2. Polymer Degradation and Mitigating Measures

Mechanical recycling of plastics implies an extrusion step, for the regranulation of the material. During this thermomechanical loading, the polymers will degrade to a certain degree (dependent on processing conditions, remaining stabilizers and sensitivity of the polymer to chain scission) [45]. This results in a lower viscosity. In practice, this translates to a higher MFI for PO or a lower IV for PET, which may change the polymer's suitability to a certain processing technique. We have elected to express the acceptable degradation per class in terms of these practical processing-related values.

Under certain circumstances, it is also possible for the viscosity of HDPE to increase rather than decrease, due to crosslinking [55]. However, under typical recycling circumstances, this is not a prevalent occurrence and we have chosen to disregard it for the elegance of the classification.

Do note that it is possible to mitigate or prevent this degradation during the recycling process. In fact, solid-state polycondensation of PET is common in most PET recycling processes [9,10], as is the addition of stabilizers to PO.

Appendix A.8.3. Contamination Levels

The effect of contaminations on recycled plastics is manifold. On a practical level, we differentiate between non-polymer contaminants (like wood, paper, dust, etc.) and contamination by polymers other than the target polymer [56].

Non-polymer contaminants will not melt during processing; the severity of their effect is dependent on the process used. Film blowing, for example, is extremely sensitive to non-melting contaminants, which will cause a tear in the fine film bubble [57]. Extrusion of bulky products on the other hand is fairly forgiving to any type of contamination, due to the large gates and bulky products themselves.

Also during the use phase of the product, non-polymer contaminants will reduce quality, as they are typically stress concentrators and will lower the functional mechanical properties [58].

To reduce the non-polymer contaminants as much as possible, a melt filtration step is typically used during the regranulation of the recycled plastics [59]. Table 1 includes threshold values for the mesh size of such melt filters. These are process dependent rather than polymer dependent.

Polymeric contaminants are further subdivided in contamination by 'similar' polymers (polyolefins in one another) and others. The latter has a more detrimental effect than the former. Contamination by other polymers can affect both the processability and the properties of the final product. Polymers do not mix in the melt phase, which will always lead to phase separation upon solidification [60], causing a complex phase morphology that is typically less ductile and less strong. This effect is typically more pronounced for chemically dissimilar polymers. Polymers with a significantly higher melt temperature than the target polymer will act similar to non-polymer contaminants, seeing as how they will not melt.

**Appendix B. Additional Results**

*Appendix B.1. Amount and Polymer Purity of Recycled Plastic from an Optimal PPW Recycling Value Chain*

The net packaging recycling rate is calculated based on the mass of the plastic packaging materials on the market and the amount of recycled plastic packaging (Equation (A1)). The number of recycled plastics from the PPW recycling value chain can be expressed in two ways. Firstly, the total amount of recycled plastics resulted from the PPW recycling value chain that also includes non-packaging plastic contributions. Secondly, the number of packaging plastics that end up in the recycled plastics from the PPW recycling value chain. The latter is used to calculate the net packaging recycling rate. This is in line with the calculation rules for the recycling targets set by the European and Dutch government [61,62].

The total amount of plastic packages that were put on the market in 2017 was 512 kton [48]. Some plastic packaging types were replaced by non-plastic packages in the model of the circular PPW recycling value chain (Appendix A.3, Table A1). The total amount of plastic packaging put on the market in a circular PPW recycling value chain was therefore estimated to be 512 kton minus the amount of replaced packaging types (3 kton). The total amount of plastic packaging waste for the PI-PWW recycling value chain was calculated by the difference between the total amount of plastic packaging put on the market and the amount of plastic packaging waste collected via the PC-PWW recycling scheme. Changes in the PC-PPW recycling value chain can thus affect the net packaging recycling rate of the PI-PPW recycling value chain.

The resulting recycled plastics (washed milled goods, WMG) and their polymeric purity of 2017 are listed in Table A6 and for the optimal PPW recycling value chain in Table A7. The amount of WMG is expressed as the total amount of product, including non-packaging plastics and other materials in the WMG. Additionally, the amount of packaging material in the WMG is calculated, to enable the calculation of the net packaging recycling rate, as described above. The polymeric purity is the percentage of targeted material in the WMG. The target material of the recycled plastic named "PO-mix" is PE and PP. Next to the polymeric purity, also the amount of black & other plastics and laminates are provided in Table A7, as this category could also include targeted materials. The extended composition of the recycled plastics in an optimal value chain is given in Table A8.

**Table A6.** The amounts of washed milled goods and their polymeric purity in 2017.

| Type of PPW | Amount of WMG [Gg] | Amount of Packaging Material in WMG [Gg] | Polymeric Purity of WMG-Target Material(s) [%] | Black & Other Plastics and Laminates Included. [%] | Quality Classification |
|---|---|---|---|---|---|
| PC PET bottles | 12.5 | 12.4 | 98.8 | 0.1 | F-NC |
| PC PE rigid | 13.1 | 12.8 | 92.6 | 0.3 | NF-SC |
| PC PP rigid | 14.9 | 12.9 | 92.0 | 2.0 | NF-SC |
| PC PE film | 23.9 | 18.3 | 82.0 | 5.6 | NF-SC |
| PC Mix (PO mix) | 38.9 | 33.9 | 87.7 | 9.6 | NF-SC |
| PC PET trays | NA | NA | NA | | |
| PC PET bottles by-product (PO mix) | 1.6 | 1.6 | 91.7 | 0.7 | NF-SC |
| PC film by-product (scenario 2) (PO mix) | NA | NA | NA | | |
| PI PET bottles deposit-refund (DR) | 19.2 | 19.2 | 99.9 | | F-NC |
| PI PET bottles DR by-product (PO mix) | 1.9 | 1.9 | 90.7 | | NF-SC |
| PI B2B | 79.9 | 79.9 | 99 | | NF-LC |
| Other PI | NA | NA | NA | | |

**Table A7.** The amounts of washed milled goods, amount of packaging materials in the washed milled goods, their polymeric purity in an optimal circular PPW recycling value chain and quality classification.

| Type of PPW | Amount of WMG [Gg] | Amount of Packaging Material in WMG [Gg] | Polymeric Purity of WMG-Target Material(s) [%] | Black & Other Plastics and Laminates Included. [%] | Quality Classification |
|---|---|---|---|---|---|
| PC PET bottles | 23.3 | 23.3 | 99.7 | 0.03 | F-NC |
| PC PE rigid | 26.4 | 26.1 | 98.1 | 0.2 | NF-LC |
| PC PP rigid | 49.3 | 47.5 | 97.7 | 0.3 | NF-LC |
| PC PE film | 49.9 | 45.7 | 98.7 | 1.0 | NF-LC |
| PC Mix (PO mix) | 32.1 | 27.8 | 93.4 | 5.5 | NF-SC |
| PC PET trays | 51.4 | 51.3 | 99.8 | 0.1 | F-NC |
| PC PET bottles by-product (PO mix) | 2.8 | 2.8 | 92.3 | 0.3 | NF-SC |
| PC film by-product (scenario 2) (PO mix) | 16.0 | 15.4 | 90.9 | 9.1 | NF-SC |
| PI PET bottles deposit-refund (DR) | 19.2 | 19.2 | 99.9 | | F-NC |
| PI PET bottles DR by-product (PO mix) | 1.9 | 1.9 | 90.7 | | NF-SC |
| PI B2B | 79.9 | 79.9 | 99 * | | NF-LC |
| Other PI | 25.5 | 25.5 | 97 * | | *NF-LC* |

*: estimation.

**Table A8.** The composition of the washed milled goods and their applicability in an optimal circular PPW recycling value chain.

| Type of PPW | PET [%] | PP [%] | PE [%] | PS [%] | PVC [%] | Paper [%] | Metal [%] | Glass [%] | Other Polymers, incl. Black [%] | Undefined, Residue, Textiles, etc. [%] | Applicability Classification [F-NC, NF-LC, NF-SC, NF-HC] |
|---|---|---|---|---|---|---|---|---|---|---|---|
| PC PET bottles | 99.7 | 0.05 | 0.1 | 0.05 | 0.0 | 0.0 | 0.0 | 0.0 | 0.0 | 0.1 | F-NC |
| PC PE rigid | 0.0 | 1.7 | 98.1 | 0.0 | 0.0 | 0.0 | 0.0 | 0.0 | 0.2 | 0.0 | NF-LC |
| PC PP rigid | 0.0 | 97.7 | 2.0 | 0.0 | 0.0 | 0.0 | 0.0 | 0.0 | 0.3 | 0.0 | NF-LC |
| PC PE film | 0.0 | 0.3 | 98.7 | 0.0 | 0.0 | 0.0 | 0.0 | 0.0 | 1.0 | 0.0 | NF-LC |
| PC Mix (PO mix) | 0.2 | 27.3 | 66.1 | 0.5 | 0.3 | 0.0 | 0.0 | 0.0 | 5.6 | 0.0 | NF-SC |
| PC PET trays | 99.8 | 0.0 | 0.0 | 0.0 | 0.0 | 0.0 | 0.1 | 0.0 | 0.1 | 0.0 | F-NC |
| PC PET bottles by-product (PO mix) | 7.4 | 13.5 | 78.7 | 0.1 | 0.0 | 0.0 | 0.0 | 0.0 | 0.3 | 0.0 | NF-SC |
| PC film by-product *(scenario 2)* (PO mix) | 0.0 | 56.7 | 34.2 | 0.0 | 0.0 | 0.0 | 0.0 | 0.0 | 9.1 | 0.0 | NF-SC |
| PI PET bottles deposit-refund (DR) | 99.9 | 0.0 | 0.1 | | | 0.0 | | | | | F-NC |
| PI PET bottles DR by-product (PO mix) | 9.3 | 23.3 | 67.4 | | | 0.0 | | | | | NF-SC |

*Appendix B.2. Autonomous Improvements in the Recycling Value Chain*

Two types of autonomous improvement options that can be implemented by individual groups of stakeholders are known to have the largest impact on the performance of the PC-PPW recycling value chain; design-for-recycling measures by the producers and improvements in the collection systems by municipalities [17]. These calculations have been repeated with the current model for PC-PPW and the current set of two improvement measures (see Sections 2.2.1 and 2.2.2). Obviously, no changes were made in the PI-PPW recycling value chain, since we cannot model these changes, yet. The result of these calculations is given in Table A9. Maximising the collection rate of the PC-PPW by the municipalities would increase the net packaging plastic recycling from 26% to 45%, without affecting the polymeric purity of the recycled plastic. Full implementation of the design-for-recycling guidelines (as described in Appendix A.3) would increase the average polymeric purity of the post-consumer recycled plastics from 91% to 95% and simultaneously improve the net packaging recycling rate from 26% to 33%.

**Table A9.** Circular performance indicators of Dutch PPW recycling value chain after two independent sets of improvement measures have been executed by two different groups of stakeholders.

| CPIs | 2017 | Maximised Packaging Collection Rate | All Packages Designed for Recycling |
|---|---|---|---|
| PC-PPW net packaging recycling rate | 26% | 45% | 33% |
| PI-PPW net packaging recycling rate | 63% | 63% | 63% |
| Total PPW net packaging recycling rate | 38% | 51% | 42% |
| PC-PPW average polymer purity | 91% | 91% | 95% |
| PI-PPW average polymer purity | 97% | 97% | 97% |
| Average polymer purity of all WMG from PPW | 93% | 93% | 95% |

*Appendix B.3. Value Chain Variations*

In Section 4.3, three different variations in the structure of the optimal circular value chain are discussed. The underlying modelling results of two of these structural variations are presented here, as the results of adding the small PET bottles to the DRS are discussed in the main text.

The first structural variation deals with sorting the flexible packages. Two scenarios are discerned. In the first scenario, the flexible packages are NIR sorted into PE flexibles, PP flexibles and a by-product of mostly laminated flexibles and missorted PE and PP flexibles. In the second scenario, the flexible packages are NIR sorted into a PE flexible and a PO-mix by-product. The results of both scenarios are given in Table A10. The consequences of adding this PO-mix by-product to the existing PC-PPW Mix is given in Table A11.

**Table A10.** Two scenarios for sorting flexible packages.

| | Amount of WMG [Gg] | Amount of Packaging Material in WMG [Gg] | Polymeric Purity of WMG-Target Material(s) [%] | Black & Other Plastics and Laminates Included. [%] |
|---|---|---|---|---|
| **Scenario 1** | | | | |
| PE film | 49.9 | 45.7 | 98.7 | 1 |
| PP film | 9.9 | 9.7 | 84.1 | 5 |
| Mix (PO mix) | 6.1 | 5.7 | 84.0 | 16 |
| **Scenario 2** | | | | |
| PE film | 49.9 | 45.7 | 98.7 | 1 |
| Mix (PO mix) | 16.0 | 15.4 | 90.9 | 9 |

**Table A11.** Polymeric purity of the PC-PPW Mix when the film sorting by-products from scenario 1 and scenario 2 are added.

| Type of PPW | Polymeric Purity of WMG-Target Material(s) [%] | Black & Other Plastics and Laminates Included. [%] |
|---|---|---|
| PC-PPW Mix (PO mix) | 93.4 | 5.5 |
| PC-PPW Mix + By-product from film sorting (PO mix) (scenario 1) | 91.9 | 7.2 |
| PC-PPW Mix + By-product from film sorting (PO mix) (scenario 2) | 92.6 | 6.7 |

The second structural aspect of the optimal circular value chain that is discussed in Section 4.3.2 of the main text is the necessity of the sorted product mixed plastics. The concomitant sorting division is listed in Table A12.

**Table A12.** Sorting division of the first sorting process of separately collected PC-PPW.

| Sorted Product (1st Sorting Process) | Sorting Division, 2017 [%] | Sorting Division, Optimal PPW Recycling Value Chain [%] |
|---|---|---|
| PET bottles | 5 | 7 |
| PET trays | 7 | 12 |
| PE rigid | 5 | 6 |
| PP rigid | 7 | 12 |
| Film | 10 | 17 |
| Mix | 26 | 12 |
| Beverage Cartons | 8 | 9 |
| Ferro metals | 6 | 7 |
| Non-ferro metals | 1 | 1 |
| Sorting residue | 22 | 14 |
| Lost moisture and dirt | 3 | 3 |

*Appendix B.4. Sensitivity Analysis*

The model was subjected to limited sensitivity analysis to understand the variation in results for one of the CPIs: the net packaging recycling rate. A sensitivity analysis with respect to the other CPI (polymeric purity) is less meaningful and more difficult to calculate since it relates strongly to the assumptions made of what the optimally designed packages are in terms of average material compositions.

The net packaging recycling rate is influenced by multiple parameters, but the net packaging collection rate proved to be the most sensitive, other parameters such as the maximal sorting fates of flexible packages during sorting and mechanical recovery proved to influence the CPI to a lesser extent. The influence of the net packaging collection rate on the net packaging recycling rate is given in Table A13. The limit was set to 70% in the model and varied for the sensitivity analysis to 60%, 80% and 90%. A 10% increase in the net collection rate of PC-PPW causes the total Dutch net packaging recycling rate to rise with 6%. The 70% limit for the net collection rate was based on empiric data of collection between 2012 and 2017 [17]. In case civilians are more encouraged to separate their packaging wastes better and restrictions are dropped, then it is imaginable that the net collection rate can increase and as a consequence the total Dutch net packaging recycling rate.

**Table A13.** Net packaging recycling rate for an optimal circular recycling value chain, with net packaging collection rate of 60, 70%, 80%, and 90%.

| CPIs | Net Collection Rate = 60% | Net Collection Rate = 70% (Limit) | Net Collection Rate = 80% | Net Collection Rate = 90% |
|---|---|---|---|---|
| PC-PPW net packaging recycling rate | 61% | 69% | 77% | 86% |
| PI-PPW net packaging recycling rate | 79% * | 78% | 78% | 78% |
| Total PPW net packaging recycling rate | 66% | 72% | 78% | 83% |

\* The total amount of plastic packaging waste for the PI-PWW recycling value chain was calculated by the difference between the total amount of plastic packaging put on the market and the amount of plastic packaging waste collected via the PC-PWW recycling scheme. Changes in the PC-PPW recycling value chain can thus slightly affect the net packaging recycling rate of the PI-PPW recycling value chain.

The influence of two other factors on the net packaging recycling rate was studied in the sensitivity analysis: the maximal sorting fates of the flexible packages and of mechanical recovery and sorting of plastic packages from municipal solid waste. Both sorting fates were independently and simultaneously raised to assess their impact on the net packaging recycling rate (Table A14).

**Table A14.** Net packaging recycling rate of the optimal PPW recycling chain, with improved sorting fates for separately collected flexible packages and improved combined fates for mechanical recovery of MSW and sorting of flexible packages.

| CPIs | Limit | Improved Wind Sifting | Higher Sorting Fates for Recovery from MSW | Improved Wind Sifting + Higher Sorting Fates for Recovery from MSW |
|---|---|---|---|---|
| PC-PPW net packaging recycling rate | 69% | 71% | 70% | 72% |
| PI-PPW net packaging recycling rate | 78% | 79% * | 79% * | 79% * |
| Total PPW net packaging recycling rate | 72% | 73% | 73% | 74% |

\* The total amount of plastic packaging waste for the PI-PWW recycling value chain was calculated by the difference between the total amount of plastic packaging put on the market and the amount of plastic packaging waste collected via the PC-PWW recycling scheme. Changes in the PC-PPW recycling value chain can thus slightly affect the net packaging recycling rate of the PI-PPW recycling value chain.

Flexible packaging plastics are sorted the least efficient of all packaging types. This inefficiency relates to the relative inefficient separation processes of ballistic separation and wind sifting. Therefore, the maximal sorting fates of flexible packages in the optimal circular PPW recycling chain were estimated to be lower than the maximal sorting fates of the rigid packages in the model of the optimal circular PPW recycling value chain. The sorting fates for flexible packages are present in two parts of the model: in the part of the model that describes the sorting of separately collected packaging waste and in the part of the model that describes the mechanical recovery and sorting of plastic packages from municipal solid waste. The original limit for the sorting fate of flexible packages that were separately collected was 80% for large flexible packages and 50% for small flexible packages. In this sensitivity analysis, both were raised to 81%. The flexible packages will undergo NIR sorting twice in the optimal recycling chain. Since the maximum sorting fate is 90% for each sorting step, the maximum overall sorting fate for both steps will be 81%. The original limit for the combined sorting fate for flexible packages that are first mechanically recovered from MSW and subsequently sorted was 45%, which was raised to 70% for the sensitivity analysis. Obviously, this raised combined sorting fate is merely an approximation of what might be possible in optimised recovery and sorting facilities. Overall, the effect of these increased sorting fates causes a rise in the net packaging recycling rate of only 1%. This proves that optimising collection is the most important method to increase the net plastic packaging recycling rate and hence also reaching compliance with the recycling targets.

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
