# Peer review of "Technical Limits in Circularity for Plastic Packages"

_sustainability, doi:10.3390/su122310021_

Round 1

Reviewer 1 Report

The research question is original and well defined. The issues addressed in the paper are of great interest to readers from different scientific fields and represent an excellent starting point for reflection on a very current topic that is the implementation of circularity concept in plastic packaging waste recycling value chain.

The paper presents an interesting input about the variations needed in the structure of a recycling value chain which wants to be more circular.

The potentiality of the proposed analysis for further researches could be more highlighted in the conclusions.

The article is written using a simple language and the use of the English language is understandable.

The paper proposes issues that are very important to analyze the actual level of implementation of circularity in a process compared to its maximum potential, so the methodology is very interesting and replicable also for other researches.

Author Response

Response to the comments made by reviewer 1

Thank you for your kind words. We agree that writing in a simple and understandable language requires more efforts than using complex sentences and idiosyncratic terminology.

When we started with this study we expected that the results of this study would predominantly be relevant for stakeholders and policy makers. But you are correct that this study will also be useful for future researchers. This study with this model under the chosen conditions (current technologies and those that are foreseen to be available within five years, focus on recycling) has given a clear result. There is a clear limit of what can be achieved; 72% recycling rate with the majority of the recycled plastics being reused in open loops. This situation requires a major disruption to progress any further. Either new technologies / processes have to be developed to enable the food grade recycling of polyolefines. Or the food contact material legislation of the EU needs to be altered in favour of food grade recycling of polyolefines. Or the industry and civilians need to move up on the waste hierarchy towards material reduction (consume less), adopt reusable packages, or develop completely new polymers that can be recycled circularly and fulfil all the functions. New research should therefore explore the potential impacts of these disruptions towards more circularity, but also research the economic and social effects. Therefore we added the following sentences on lines: 450-452 “Future research would be welcomed on the acceptability of the required efforts by the stakeholders in relation to the circularity that can be achieved. Additionally, the impact of new disruptive technologies and policies (reusable packages) on the whole value chain could be explored.”

Reviewer 2 Report

Overall comment

This study aims to explore the theoretical limits of a circular recycling value chain for packaging waste (PPW) in the Netherlands with the currently available technologies and those that are foreseen to be available within the coming five years.

To pursue such goal, the authors tries to define a set of measures, by increasing (mainly) two Circular Performance Indicators (CPIs) to their theoretical limits, namely: net packaging recycling rate and average polymer purity of the recycled plastics.

Such optimization is performed through the increase of the performance of the recycling value chain, by acting on its four stages, namely: packaging design, collection, sorting and recycling.

The authors also highlight  the importance of maximizing these two indicators, by referring the EU’s goals, and mainly, the increase in terms of availability of material (plastic) for demanding applications (e.g. food packaging).

The authors also propose a classification of recycled plastics to define their applicability, which they use as a method to predict the level of circularity that can be attained.

Although the exception of a literature review section, the paper seems to be well-organized, containing all the expected components, namely the   Introduction, Research Methods, Discussion and Analysis of results, and Conclusions.

Although the authors compares the results with other studies, a literature review section should also be included here, regarding the last approaches, and concerning the use of Circular Performance Indicators, in order to highlight  the importance of the study developed here, to the circular economy in general.

The author’s results are somehow convincing, given the purpose of the work.

However, I still reluctant to consider the contribution of this paper on behalf of sustainability or sustainable development, since that only the  technical effects are explored, through the application of different measures to achieve several scenarios here.

Sustainability has to do with sustainable measures, that allows to establish a balance between economic, social and environment dimensions. Besides the quality of this study and its importance for the circular economy of plastic packages, the economic dimension is missing and the contribution for the social dimension it is not clear for me. I think this paper would be more suitable to a journal that leads with ecological issues, given the importance of the results achieved here.

Moreover, the abstract, it’s not clear about the research method used. We only get to know more about this issue, further  on Section 2.

However, and in general, the authors have answered to the research question stated here.

Furthermore, the relevance of the subject is also updated, although the lack of (some) novelty here.

Some recommendations regarding this issue, can be found it below.

Some recommendations of improvement:

  Strong points:

  • The relevance of the subject to achieve sustainable measures, regarding these materials
  • Data used
  • Research method
  • Discussion of results: Although it should be improved (if it is possible) by discussing the obtained results by stablishing a benchmark with other similar approaches, based on the limitations of this study and the ones regarding the other studies, in order to be compared.
  • The proposal of a classification method, regarding the recycled plastics to define their applicability to predict the level of circularity

Weak points:

  • The relationship with sustainability or sustainable development: Besides the economic issue, already referred here, the contribution for the “social dimension” of sustainability is not clear for me in this work, since that what is the main contribution (for instance) for the consumer/industrial wellbeing?
  • Future work - Regarding the conclusions’ section, and despite the main research question, pointed and answered, based on the achieved results, the authors should better explore the “future work” on the same section, by pointing (for instance) some clues to a reader, who might want to pursue the research. One example, is the recommendation of future work based on model limitations, expressed on Section 4.4.

Author Response

Reviewer 2, point 1

This study aims to explore the theoretical limits of a circular recycling value chain for packaging waste (PPW) in the Netherlands with the currently available technologies and those that are foreseen to be available within the coming five years.

To pursue such goal, the authors tries to define a set of measures, by increasing (mainly) two Circular Performance Indicators (CPIs) to their theoretical limits, namely: net packaging recycling rate and average polymer purity of the recycled plastics. Such optimization is performed through the increase of the performance of the recycling value chain, by acting on its four stages, namely: packaging design, collection, sorting and recycling. The authors also highlight  the importance of maximizing these two indicators, by referring the EU’s goals, and mainly, the increase in terms of availability of material (plastic) for demanding applications (e.g. food packaging). The authors also propose a classification of recycled plastics to define their applicability, which they use as a method to predict the level of circularity that can be attained. Although the exception of a literature review section, the paper seems to be well-organized, containing all the expected components, namely the Introduction, Research Methods, Discussion and Analysis of results, and Conclusions. Although the authors compares the results with other studies, a literature review section should also be included here, regarding the last approaches, and concerning the use of Circular Performance Indicators, in order to highlight  the importance of the study developed here, to the circular economy in general.

Response to reviewer 2, comment 1

Thank you for your correct point. Indeed we have to explain why we use these two circular performance indicators and not any of the hundreds others that either have been used or have been proposed. A review of all circular performance indicators would, however, be a topic for a review paper itself. But fortunately, such a review have been written and indeed we should refer to it. Consequently, we added the following sentences on lines 62-73 of the introduction: “Circular indicators assess the level of circularity that a product, company or collection and recycling network has achieved. Hundreds of these indicators have previously been proposed and used [20-22]. Specific for PPW, simple recycling rates are insufficient since these do not account for the quality of the recycled plastic and whether or not the material is kept within material circles [23]. As possible solutions closed loop recycling rates and open loop recycling rates have been proposed [23], but also quality factors in which the type of material circulation is accounted for [24]. In this study we will use two CPI’s that result from the material flow analysis. These are the net packaging recycling rate and the (average) polymeric purity of the recycled plastics produced. The polymeric purity relates to the applicability of the recycled plastic and hence to the type of material cycles the recycled plastics are used in.”

Reviewer 2, point 2

The author’s results are somehow convincing, given the purpose of the work. However, I still reluctant to consider the contribution of this paper on behalf of sustainability or sustainable development, since that only the  technical effects are explored, through the application of different measures to achieve several scenarios here. Sustainability has to do with sustainable measures, that allows to establish a balance between economic, social and environment dimensions. Besides the quality of this study and its importance for the circular economy of plastic packages, the economic dimension is missing and the contribution for the social dimension it is not clear for me. I think this paper would be more suitable to a journal that leads with ecological issues, given the importance of the results achieved here.

Response to reviewer 2, comment 2

Thank you for raising this subject. Indeed this is a technical study and not an economic study or a social study. This is already mentioned in the title: “technical limits” and hence this study does not deal with economic or social limits. We will clarify further in the introduction that this is a technical study and not an economic or social study. Therefore we added a few words to the sentence in line 78-79: “In this theoretical study only technical argumentation is used and economic and social considerations and interrelations are ignored.” This doesn’t imply that we don’t find these other limits interesting, they are. And obviously, these domains can influence each other. But this paper answers the question what the technical limit is that could be achieved if all stakeholders would do their utmost best in a concerted action with the technology that is available within the coming five years’ time. That is a legitimate research question that is fairly complicated to answer in itself. More complex research questions can be answered in the future, when also other dimensions of sustainability are considered.

Two relationships between this technical study and social science are eminent within collection and packaging design.

Collection. The maximum net separation rate for plastic packaging waste by Dutch consumers has been studied and amounted roughly 70% for civilians participating in separate collection of plastic packaging waste. Hence this number was used as maximum separation rate and we used higher numbers in our sensitivity analysis. We are completely aware that the social science & environmental psychology has produced a large number of papers on the intrinsic motivation, biospheric values, context factors that influence the separation rate of individuals. We respect those contributions, but they do not yield more information for a technical study than that the maximum net separation rate is 70%.

Packaging design. In our model we changed the design of packages to render them better recyclable on the level of packaging types. Further improvements can be made in the future by minimising prints and convenience options (hand pumps, reclosable strips, spray guns, etc.). Then the social question becomes urgent: to what extent are civilians willing to sacrifice these options for improved recyclability? Those questions have to the best of our knowledge not been studied yet.

Reviewer 2, point 3

Moreover, the abstract, it’s not clear about the research method used. We only get to know more about this issue, further  on Section 2.

Response to reviewer 2, comment 3

You are correct that we did not mentioned the research method in the abstract, since it is fairly complicated modelling method based on material flow analysis we use. However, you are correct that we should shortly explain it. Hence we changed the abstract in the following manner. One sentence was added on line 17: “The effects of the measures were modelled with material flow analysis.”

Reviewer 2, point 4

However, and in general, the authors have answered to the research question stated here. Furthermore, the relevance of the subject is also updated, although the lack of (some) novelty here. Some recommendations regarding this issue, can be found it below. Some recommendations of improvement:

Strong points:

The relevance of the subject to achieve sustainable measures, regarding these materials

Data used

Research method

Discussion of results: Although it should be improved (if it is possible) by discussing the obtained results by stablishing a benchmark with other similar approaches, based on the limitations of this study and the ones regarding the other studies, in order to be compared.

The proposal of a classification method, regarding the recycled plastics to define their applicability to predict the level of circularity

Weak points:

The relationship with sustainability or sustainable development: Besides the economic issue, already referred here, the contribution for the “social dimension” of sustainability is not clear for me in this work, since that what is the main contribution (for instance) for the consumer/industrial wellbeing?

Future work - Regarding the conclusions’ section, and despite the main research question, pointed and answered, based on the achieved results, the authors should better explore the “future work” on the same section, by pointing (for instance) some clues to a reader, who might want to pursue the research. One example, is the recommendation of future work based on model limitations, expressed on Section 4.4.

Response to reviewer 2, comment 4

Regarding your point to discuss our results with those of other studies, we would happily do that in case there would have been previous related technical studies. But unfortunately, there are no other evidence-based technical studies that explore the maximum levels of circularity for plastic packages that can be attained in a modern society. This is not surprising, as the models required to make those calculations are complex and require substantial databases that are seldom available.

The relation with social science has already been answered in comment 3.

The relation to future work has already been discussed in response to the first reviewer, namely:

When we started with this study we expected that the results of this study would predominantly be relevant for stakeholders and policy makers. But you are correct that this study will also be useful for future researchers. This study with this model under the chosen conditions (current technologies and those that are foreseen to be available within five years, focus on recycling) has given a clear result. There is a clear limit of what can be achieved; 72% recycling rate with the majority of the recycled plastics being reused in open loops. This situation requires a major disruption to progress any further. Either new technologies / processes have to be developed to enable the food grade recycling of polyolefines. Or the food contact material legislation of the EU needs to be altered in favour of food grade recycling of polyolefines. Or the industry and civilians need to move up on the waste hierarchy towards material reduction (consume less), adopt reusable packages, or develop completely new polymers that can be recycled circularly and fulfil all the functions. New research should therefore explore the potential impacts of these disruptions towards more circularity, but also research the economic and social effects. Therefore we added the following sentences on lines: 450-452 “Future research would be welcomed on the acceptability of the required efforts by the stakeholders in relation to the circularity that can be achieved. Additionally, the impact of new disruptive technologies and policies (reusable packages) on the whole value chain could be explored.”

We hope that this has sufficiently answered the comments given and are looking forward to your response.

Round 2

Reviewer 2 Report

The authors have answered to my questions.